# Assessing Walking Stability Based on Whole-Body Movement Derived from a Depth-Sensing Camera

**DOI:** 10.3390/s22197542

**Published:** 2022-10-05

**Authors:** Arunee Promsri

**Affiliations:** 1Department of Physical Therapy, School of Allied Health Sciences, University of Phayao, 19 Moo 2, Maeka, Muang, Phayao 56000, Thailand; arunee.pr@up.ac.th; 2Unit of Excellence in Neuromechanics, School of Allied Health Sciences, University of Phayao, 19 Moo 2, Maeka, Muang, Phayao 56000, Thailand

**Keywords:** gait, variability, neuromuscular control, sex difference, movement structure, movement synergy, treadmill walking, depth camera, Kinect, principal component analysis (PCA)

## Abstract

Stability during walking is considered a crucial aspect of assessing gait ability. The current study aimed to assess walking stability by applying principal component analysis (PCA) to decompose three-dimensional (3D) whole-body kinematic data of 104 healthy young adults (21.9 ± 3.5 years, 54 females) derived from a depth-sensing camera into a set of movement components/synergies called “principal movements” (PMs), forming together to achieve the task goal. The effect of sex as the focus area was tested on three PCA-based variables computed for each PM: the relative explained variance (rVAR) as a measure of the composition of movement structures; the largest Lyapunov exponent (LyE) as a measure of variability; and the number of zero-crossings (N) as a measure of the tightness of neuromuscular control. The results show that the sex effects appear in the specific PMs. Specifically, in PM_1_, resembling the swing-phase movement, females have greater LyE (*p* = 0.013) and N (*p* = 0.017) values than males. Moreover, in PM_3_, representing the mid-stance-phase movement, females have smaller rVAR (*p* = 0.020) but greater N (*p* = 0.008) values than males. These empirical findings suggest that the inherent sex differences in walking stability should be considered in assessing and training locomotion.

## 1. Introduction

Walking is one of our daily functional motor tasks. It is accepted that the ability to walk effectively and securely on level or uneven terrains is essential for maintaining independence and reflects physical performance for preventing falls [1]. In this sense, walking stability, which is typically defined as the ability of a system (e.g., the volitional movement system) to maintain its original state in the face of internal (e.g., neuromuscular) and external (e.g., environmental) disturbances, has been considered one of the essential aspects when evaluating gait ability [2,3]. Since stability parameters offer information about the noise inherent in motor task performance and explicitly quantify the performance of dynamic error correction, assessing the variability has also been used as one indirect measure of a person’s walking stability [2,3]. The non-linear methods, e.g., local dynamic stability quantified by the determination of the largest Lyapunov exponent (LyE), have widely been applied to analyze stability during locomotion in terms of assessing the ability of the neuromuscular system to attenuate for small internal or external perturbations in order to maintain functional locomotion, with a greater LyE indicating more variability or lesser stability [3,4,5].

Regarding the numerous redundant degrees of freedom in the motor apparatus [6], it is believed that the central nervous system (CNS) finds a near-optimal solution to govern the given human movements via task-relevant synergistic muscle activations [7]. Hence, when observing the motor behaviors, a combination of different task-dependent movement synergies (i.e., movement patterns) working together to achieve the given motor task goal is revealed [8,9], of which these movement synergies can be adapted to internal and external demands [10]. The applications of dimensionality reduction techniques to three-dimensional (3D) kinematic data have recently been applied to gain information on movement synergies by reducing the number of features required to complete the task by creating fewer new variables and retaining the most information about how individuals move from the original feature set [8,9,11]. For instance, principal component analysis (PCA) is applied to the individual multi-segment movements typically recorded from a full-body marker set and tracked by the infrared cameras of the motion tracking system into a set of one-dimensional movement components/synergies called “principal movements” (PM_k_), where *k* is the order of movement components [8,9]. Individual PMs can be visualized as different movement synergies, resembling the swing phase, stance phase, and other relevant movements, cooperatively forming together to complete the walking tasks [12,13]. PCA-based posture space also provides data on each PM’s position and acceleration [9], reflecting the neuromuscular control of each PM_k_ in terms of system forces [8,10] and myoelectric activities [14]. Furthermore, in order to assess the local dynamic stability during gait, LyE has been applied to individual PMs’ positions, reflecting the inherent ability of the neuromuscular system to control infinitesimal perturbations of individual movement components [13,15].

When considering the sex-specific walking characteristics, it is thought that males and females walk differently and have distinct kinematic gait characteristics, of which the anthropometric differences between the sexes are accepted as one potential cause of the sex differences in walking characteristics [16,17]. Specifically, males have longer stride lengths and faster gait speeds at a self-selected walking speed due to their larger stature than females [16]. Moreover, investigating the walking differences between the sexes is of interest for various clinical applications. For example, sex differences in kinematic gait characteristics exist in individuals with several types of pathologies, e.g., osteoarthritis [18,19], iliotibial band syndrome [20], and Down’s syndrome [21]. Hence, analyzing the stability of the main movement synergies decomposed from the whole-body walking movements may beneficially provide an understanding of the inherent neuromuscular control involved in individual gait patterns, since the neuromuscular system controls posture through muscles that generate relative movements between multiple segments of the body [22].

Practically, although the 3D motion analysis systems based on the reflective marker recording are capable of measuring whole-body movements and producing highly accurate and precise results as a gold standard, the high cost, lengthy preparation time, and need for specialized staff to operate these systems are barriers to their widespread adoption, such as in routine clinical care [23]. Alternatively, taking measurements outside the laboratory, such as with a low-cost, depth-sensing camera (e.g., Microsoft Kinect, Microsoft, USA), provides a portable and cost-effective markerless 3D motion capture device that enables depth evaluation to be integrated with typical 2D photographs [24]. In addition, the Microsoft Kinect has been suggested as a potential alternative tool with relatively high reliability and validity for clinical gait analysis [24,25,26,27,28].

In summary, the main purpose of the current study was to investigate walking stability in terms of the neuromuscular control of the main movement synergies forming together to achieve the treadmill walking task in healthy young adults, in which a sex difference was a focus area. PCA was used to characterize movement components/synergies (i.e., principal movements, PMs) from the 3D whole-body movements derived from a depth-sensing camera. PCA-based variables were then determined to differentiate the sex differences in walking stability. As previously reported [16,17], males and females have distinct kinematic gait characteristics. Therefore, it was hypothesized that the differences between the sexes in walking stability would manifest in the specific PMs related to the specific gait patterns.

## 2. Materials and Methods

### 2.1. Secondary Data Analysis

The kinematic datasets of the whole-body treadmill walking movements retrieved from 104 healthy young participants (50 males and 54 females) for the current investigation were obtained from a peer-reviewed open-access dataset [29]. Table 1 represents the characteristics of the participants.

The measurement procedures were fully described in Guffanti et al. [29]. In brief, whole-body movement was recorded using a single Microsoft Kinect V2 depth sensor (Microsoft Corporation, Redmond, WA, USA) placed in frontal view for 1.8 m in front of a 1.2 m-long treadmill. The speed of the motorized treadmill gradually increased from 0 m/s to 1.2 m/s, and all the recordings started once the 1.2 m/s speed was attained. Then, the recording stopped after 30 s of continuous walking, and slowdown started. A systematic review reported that walking at 1.2 m/s is commonly considered a slow-to-moderate speed for healthy young adults [30].

The Kinect V2 for Windows Software Development Kit (SDK 2.0, Microsoft Corporation, Redmond, WA, USA) provides, with a sampling rate of 30 Hz, the three-dimensional (3D) positions of 25 body parts, including head, neck, spine shoulder, spine mid, spine base, right and left shoulder, elbow, wrist, hand, thumb, hand tip, hip, knee, ankle, and foot. However, although the Kinect V2 camera has a target sampling rate of 30 Hz, the frequency was not constant and could not be stabilized. The actual sampling rate recorded in this investigation was 34.4 ± 4.3 Hz, consistent with a prior study [30,31]. Thus, the sampling frequency of each dataset was resampled to 30 Hz before further analysis.

### 2.2. Movement Synergy Extraction

All data processing was performed in MATLAB version 2021b (MathWorks Inc., Natick, MA, USA). Movement synergies were quantified using a kinematic principal component analysis (PCA) [32]. The matrix of a whole-body walking movement of each participant contained 25 markers (x, y, z) corresponding to 75 spatial coordinates (i.e., a 75-dimensional posture vector-matrix). An example of whole-body walking movement received from one male participant can be viewed in an animated stick figure video (Appendix A).

Then, three steps of kinematic data preprocessing were conducted, of which the mathematic-detailed procedures have been fully described in previous studies [33,34]. First, the matrices of all participants were individually centered by subtracting the mean posture vector to eliminate differences in mean marker positioning in space from influencing the PCA outcomes [8,35]. Second, the centered posture vectors were normalized using the mean Euclidean distance to address anthropometric differences [8,30]. Third, the normalized posture vectors were weighted by considering sex-specific mass distributions of each body segment to eliminate the inherent differences in body size between males and females [36], allowing for assessing the sex differences in PCA-based variables [15,32,37]. Finally, the weighted posture vectors of all participants were concatenated to form a 93,600 × 75-input matrix (30 [sampling rate (Hz)] × 30 [testing duration (s)] × 104 [number of participants] x 75 [marker coordinates]) for the PCA.

PCA was computed by performing a singular value decomposition on the covariance matrix of the input matrix through a public “PManalyzer” software [9], yielding a set of PC_k_ vectors, eigenvalues, and scores. Each PC vector defined a specific pattern of correlated marker movements called “*principal movements*” (PMs) [8], from which animated stick figures created from each PC_k_ vector can be produced as a visual representation of each PM, resembling different movement synergies forming together to achieve the task goal. The eigenvalues quantify the contribution of the associated PM_k_ to the overall variance, i.e., how much (in percent) variance there is in individual PMs. In addition, the scores are obtained by projecting the normalized posture vectors onto the PC_k_ vectors, representing the actual postural movements expressed in the coordinates defined by the PC_k_ vector basis. Thus, the resultant PC_k_ scores can be interpreted as “*principal positions*”, or PP_k_(*t*), where (*t*) denotes that these variables are functions of time *t* [38]. In analogy with Newton’s mechanics, the “*principal accelerations*” (PA_k_(*t*)) can be obtained by differentiating PP_k_(*t*) twice. PA_k_(*t*) are reported to be associated with the lower-limb electromyographic (EMG) data in postural control tasks [14], supporting their effectiveness in assessing neuromuscular control of individual PMs not only observed in balance tasks [10,32,33,35,39,40], but also in other types of human movement tasks (e.g., walking [12,41] and running [15]). The word “*principal*” in the titles of the kinematic variables indicates that these variables were obtained through PCA [9].

In order to avoid noise amplification in the differentiation processes, a Fourier analysis was conducted on the raw PP_k_(t) [34], revealing that the highest power resided in frequencies around 2–4 Hz, but visible power was still observed in the frequency range between 5 and 8 Hz. Therefore, the time series were filtered with a third-order zero-phase 8 Hz low-pass Butterworth filter before performing the differentiation step. In addition, leave-one-out cross-validation was used to determine the vulnerability of individual PM_k_ and the PCA-based dependent variables that change the input data matrix in order to address validity considerations [9]. The first five PCs proved robust, explained 91.9% of the total relative explained variance, and were selected to test the hypotheses.

### 2.3. Independent Variable Computation

In order to determine the sex differences in the walking variability, three PCA-based variables were computed for each participant and each PM. First, the subject-specific *relative explained variance* of PP_k_(*t*) or PP_k__rVAR was calculated to determine how much (in percent) the contribution of each PM to the total variance in postural positions was [8,35]. In this sense, the rVAR_k_ quantifies how important each PM_k_ is for the overall coordinative movement structures of the treadmill walking movements. If there are differences in PP_k__rVAR between males and females, they indicate differences in the coordinative structure of the overall movements.

Second, in order to quantify walking variability, a non-linear method, the subject-specific *largest Lyapunov exponent* (LyE) of PP_k_(*t*) or PP_k__LyE [13], was computed to assess the local dynamic stability, i.e., the ability of the neuromuscular system to attenuate small perturbations to maintain functional locomotion seen as the divergent trajectories in state space, i.e., the temporal variability of gait [5,13,15]. LyE was calculated by applying Wolf’s algorithm [42], of which the time delay (*τ* = 9) [43] and embedding dimension (m = 4) [43] were determined using the average mutual information (AMI) [44] and the false nearest neighbor algorithms [45], respectively. A greater LyE value indicates an inability of the neuromuscular system to diminish the perturbations, resulting in a higher divergence of the state space trajectories that reflects the lower individual’s walking stability [5,13].

Third, the subject-specific *number of zero-crossings* of PA_k_(*t*) or PA_k__N was computed as a measure of how often the whole-body acceleration (i.e., the neuromuscular control system) changed the direction of its intervention (i.e., the tightness of the neuromuscular control in each PM), with higher PA_k__N representing tighter control of the given PM_k_ than lower PA_k__N [10,12,35]. In other words, PA_k_(t) values cross zero whenever the direction of the acceleration changes, i.e., when the neuromuscular system counteracts the movement accelerations; hence, the PA_k__N serves as a measure of how tightly the neuromuscular system controls the motion of individual movement components [9].

### 2.4. Statistical Analysis

All statistical analyses were conducted using the IBM SPSS Statistics software version 26.0 (SPSS Inc., Chicago, IL, USA), with the alpha level set at a = 0.05. Kolmogorov–Smirnov tests suggested independent-sample t-tests for testing sex differences, of which Cohen’s *d* was computed as the effect size [46]. However, the variable PP_k__rVAR was not normally distributed. Hence, the corresponding nonparametric test, the Mann–Whitney U test, was conducted in these cases, and Rosenthal’s *r* was used as the effect size [47].

## 3. Results

Table 2 represents the descriptive movement characteristics of the first five movement components (PM_1−5_), of which the visualizations of PM_1−5_ are depicted in Figure 1. The first movement component (PM_1_) resembled the swing-phase movement of the gait cycle. The second and third movement components (PM_2__−3_) are associated with the single-limb support phase, whereas the fourth and fifth movement components (PM_4__−5_) are mainly related to the weight acceptance of the stance phase of the gait cycle. Movement components are more precise and can be easily characterized when viewed in an animated stick figure video (Appendix A).

As represented in Table 3, the main findings show that the sex differences in walking stability assessed through three PCA-based variables (PP_k__rVAR, PP_k__LyE, and PA_k__N) are observed in the specific PMs. Specifically, in PM_1_, resembling the swing-phase movement component, females have more variability (PP_1__LyE; (*p* = 0.013; (small effect size) *d* = 0.484)) and more tightness of neuromuscular control (PA_1__N; (*p* = 0.017; (small effect size) *d* = 0.483)) of this movement component than males. In addition, in PM_3_, representing the mid-stance-phase movement component, females have a smaller contribution (PP_3__rVAR (*p* = 0.020; (small effect size) *r* = −0.465)), but a greater tightness of neuromuscular control (PA_3__N (*p* = 0.008; (medium effect size) *d* = 0.548)) of this movement component than males.

## 4. Discussion

The current study investigated the effects of sex on walking stability in healthy young adults by analyzing the movement synergies, i.e., principal movements (PMs) of whole-body treadmill walking movements, which were decomposed using principal component analysis (PCA). The sex differences as the focus area were tested on three PCA-based variables—PP_rVAR, PP_LyE, and PA_N—to quantify the composition of movement structures, variability, and tightness of the neuromuscular control of each PM. In agreement with the hypotheses, the main findings show that the differences in walking stability between males and females appear in the specific PMs, mainly representing the swing phase (PM_1_) and the mid-stance phase (PM_3_) of the gait cycle [48].

As reported in traditional gait analysis [16], healthy young females have shorter stride lengths and slower gait speeds at a self-selected walking speed than healthy young males, possibly due to their shorter height. In addition, females exhibited more pelvic movement obliquity, arms swinging, torso rigidity, and attenuated accelerations from the pelvis to the head than males [17]. Contrary to conventional analysis, assessing the gait patterns from a whole-body segment movement revealed the main movement strategies linked to the stance and swing phases formed together to move the body forward [12,13]. The higher variability in the swing phase and tighter neuromuscular control of both the stance and swing phases observed in females compared to males agreed with the impacts of sex difference on gait [16,17]. These findings may benefit injury prevention and rehabilitation in locomotion, as sex differences have been identified as a possible injury risk factor, with females being the most affected [49].

In particular, when considering the swing phase (PM_1_), females have greater temporal variability (PP_1__LyE) and tighter control (PA_1__N) than males. Typically, the goals of the swing phase of the gait cycle are to clear the foot off the ground, conduct a forward swing of the limb, and prepare the stance for the next step [48]. Moreover, LyE estimates the rate of divergence of close trajectories in state space by measuring the rate at which the waveform shape of a time series varies from step cycle to step cycle [4,13]. Therefore, a possible interpretation of the greater temporal variability seen in females might be that females exhibited less dynamic or local stability in the swing phase movement strategy (PM_1_) than males. Moreover, when focusing on the features of the mid-stance phase (PM_3_), the mid-stance is the first sub-phase of single-limb support in which the lower leg rotates forward over the supporting foot, creating the rocker motion of the cycle to maintain the forward progression of gait [48]. In addition, the pre-swing is a transitional period between the stance and swing in which the foot pushes off the ground and lifts off [48]. Since PM accelerations correlate with lower-limb electromyographic activity [14], the tighter neuromuscular control (PA_3__N) observed in females may be interpreted as a sign of how neuromuscular control differs between sexes. In other words, females may need tighter neuromuscular control to maintain weight-bearing stability and move from the stance phase to the swing phase, since the single-limb support of the stance phase involves progressive body movement over the foot’s weight-bearing stability [48].

In terms of practical application, the experimental setup of the Microsoft Kinect provides essential insights into changes in gait performance [30], which can be transferred to a real-world scenario because there is no need for time-consuming marker placement, and the system can be operated in a community or different environments. However, one main limitation is that the Kinect V2 kinematic data are not sampled consistently at 30 Hz throughout all studies, requiring resampling the signal at a constant sampling frequency before data processing [30]. For future research, a combination study of muscle activity and self-specific manners related to walking stability is of interest, since the strength of correlation level between myoelectric activities and principal accelerations (PA_k_) differs between muscles, indicating that specific muscles play an important specific role in specific movement components [14]. Moreover, it may be of interest to focus on the effects of walking speed and external perturbation on walking stability.

## 5. Conclusions

The current study decomposed the movement components/synergies (i.e., principal movements, PMs) of the whole-body treadmill walking movements derived from a depth-sensing camera (Kinect V2). The effects of sex as the focus area were observed in the specific main movement strategies, resembling the swing and mid-stance phases of the gait cycle. Specifically, in the swing-phase movement component (PM_1_), females have greater variability and more tightness of neuromuscular control than males. In addition, females also have a greater contribution to the mid-stance-phase movement component (PM_3_), with more tightness of neuromuscular control of this movement synergy than males. The findings suggest that the inherent sex differences in walking stability should be considered one aspect while assessing or training the locomotion.

## Figures and Tables

**Figure 1 sensors-22-07542-f001:**
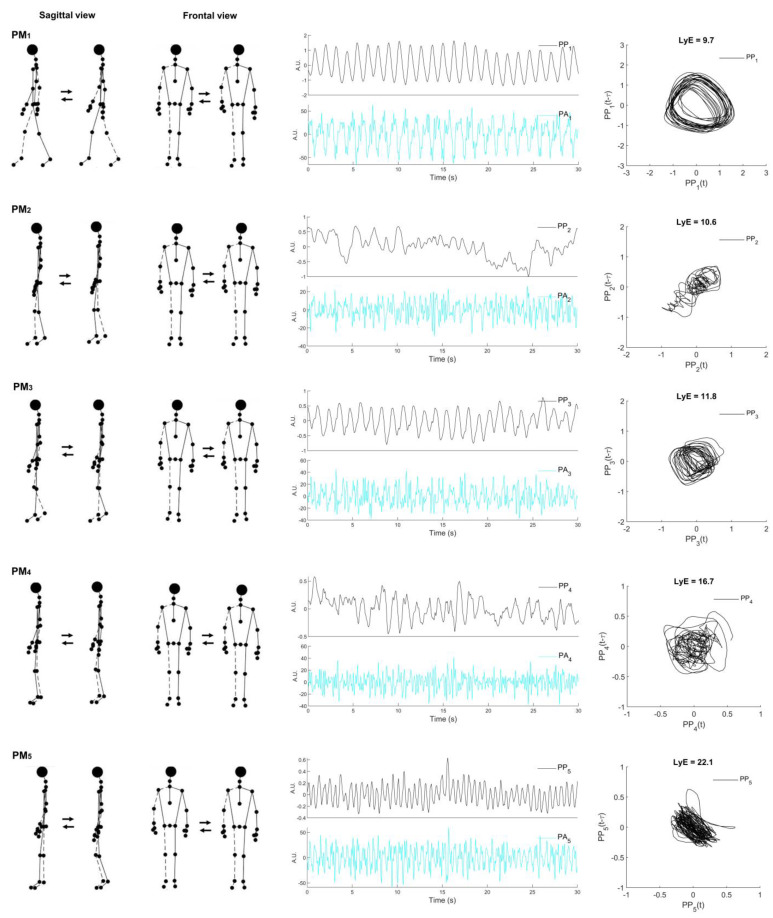
Visualized representation of the first five principal movements (PM_1–5_; **left column**) determined from treadmill walking at 1.2 m/s. Example data of principal position (PP_k_) and principal acceleration (PA_k_) (**middle column**) over time and the space-time representation for calculated Lyapunov exponent (LyE) of individual PP_k_ (**right column**) were derived from one male participant, *k* denoted the order of principal components. Note: the dashed line indicates the right limb.

**Table 1 sensors-22-07542-t001:** Characteristics of participants (* *p* < 0.001).

	Male (n = 50)	Female (n = 54)	*p*-Value
Age (years)	22.4 ± 3.4	21.3 ± 3.5	0.124
Weight (kg)	71.6 ± 9.2	61.6 ± 10.0	<0.001 *
Height (cm)	177.6 ± 6.3	165.3 ± 6.1	<0.001 *
Body mass index (kg/m^2^)	22.7 ± 2.6	22.5 ± 3.2	0.091

**Table 2 sensors-22-07542-t002:** The relative explained variance of PP_k_ (PP_k__rVAR) and descriptive movements of the first five principal movements (PM_1−5_) analyzed from treadmill walking.

PM_k_	PP_k__rVAR (%)	Descriptive Movements
1	53.3 ± 9.5	The swing phase: anti-phase arm and leg movements in the sagittal plane
2	19.7 ± 9.0	The single-limb support phase closely related to the terminal stance phase: anti-phase hip flexion and extension movements
3	9.9 ± 2.6	The single-limb support phase closely related to the mid-stance phase: anti-phase knee flexion and extension movements combined with a lateral shift of the upper body onto the stance leg
4	6.7 ± 3.1	Weight acceptance of the stance phase: lateral weight shift with small knee flexion posture
5	2.3 ± 1.1	Weight acceptance of the stance phase: knee flexion/extension movements and vertical whole-body movements combined with the sliding of the treadmill

**Table 3 sensors-22-07542-t003:** Comparisons of the relative explained variance of PP_k_ (PP_k__rVAR), the Lyapunov exponent of PP_k_ (PP_k__LyE), and the number of zero-crossing of PA_k_ (PA_k__N) of the first five principal movements (PM_1−5_) between males and females (mean ± SD; * *p* < 0.05; *p*-values smaller than 0.05 are printed in bold).

**PP_k__rVAR**	**Male**	**Female**	***p*-Value**	**Effect Size**	**Observed Power**
1	55.0 ± 9.4	51.8 ± 9.4	0.064	−0.371	0.753
2	18.8 ± 7.9	20.6 ± 9.8	0.359	−0.183	0.582
3	10.4 ± 2.6	9.3 ± 2.3	**0.020 ***	**−0.465**	**0.827**
4	6.4 ± 3.2	7.0 ± 2.9	0.079	−0.351	0.736
5	2.1 ± 0.5	2.5 ± 1.4	0.080	−0.350	0.735
**PP_k__LyE**	**Male**	**Female**	***p*-Value**	**Effect size**	**Observed power**
1	9.1 ± 1.7	10.0 ± 2.0	**0.013 ***	**0.484**	**0.847**
2	9.8 ± 2.1	9.6 ± 2.7	0.348	0.082	0.518
3	10.8 ± 2.2	11.5 ± 1.8	0.086	0.367	0.757
4	13.9 ± 2.2	14.0 ± 2.3	0.393	0.133	0.547
5	17.5 ± 1.8	18.1 ± 1.5	0.089	0.362	0.753
**PA_k__N**	**Male**	**Female**	***p*-Value**	**Effect size**	**Observed power**
1	124.4 ± 34.9	140.0 ± 30.9	**0.017 ***	**0.483**	**0.846**
2	155.4 ± 22.7	156.6 ± 26.4	0.810	0.079	0.517
3	167.6 ± 32.0	183.1 ± 25.9	**0.008 ***	**0.548**	**0.884**
4	193.7 ± 24.9	202.9 ± 27.1	0.075	0.345	0.737
5	182.3 ± 22.5	187.6 ± 24.8	0.260	0.212	0.611

## Data Availability

Not applicable.

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
