# Peer review of "Assessing Walking Stability Based on Whole-Body Movement Derived from a Depth-Sensing Camera"

_sensors, 2022, doi:10.3390/s22197542_

Round 1

Reviewer 1 Report

The author presented an analysis of the stability of gait using motion reconstruction derived from an RGB-D sensor. She proposes to use PCA to obtain principal movements and scores relative to their impact on the postural variance and the components of the posture on the PC vector basis. They then double differentiate the PC scores to obtain PAs and compute three performance indicators regarding i) the contribution of PMs to the postural variance; ii) the disturbance rejection capability of the subjects; iii) the tightness of the postural control. The authors conclude that there are differences in postural control and posture variability between males and females.

The paper is clear and the study is interesting. I have concerns regarding the methodology:

1)The correlation between EMG and PAs has been shown by the author in previous work on a different task, how can that finding be generalized?

2)The scores rely on the calculation from double differentiation of data gathered from the Kinect V2 sensor. In the previous work [12] motion data were gathered at 250Hz and then filtered at 10Hz, then double differentiated. Are kinematic data gathered in this new way comparable with the ones of the previous study?

3) There are many normalization passages that may change the differences between genders. More details about mass distribution normalization should be provided

Author Response

Response to Reviewer 1 Comments

Point 1: The author presented an analysis of the stability of gait using motion reconstruction derived from an RGB-D sensor. She proposes to use PCA to obtain principal movements and scores relative to their impact on the postural variance and the components of the posture on the PC vector basis. They then double differentiate the PC scores to obtain PAs and compute three performance indicators regarding i) the contribution of PMs to the postural variance; ii) the disturbance rejection capability of the subjects; iii) the tightness of the postural control. The authors conclude that there are differences in postural control and posture variability between males and females.

The paper is clear, and the study is interesting. I have concerns regarding the methodology:

Response 1: Thank you very much for your valuable time reviewing the manuscript and providing constructive comments and valuable suggestions. I appreciate all the suggestions and applied them to improve the study. Please find all the changes in the revised manuscript and briefly in the point-by-point responses below.

Point 2: 1) The correlation between EMG and PAs has been shown by the author in previous work on a different task, how can that finding be generalized?

Response 2: Regarding this concern, I rewrote the related sentence in the Introduction and added more information to the Method part. Based on my previous works focusing on postural control tasks, the correlation between the EMG and PA signals was observed, of which different muscles showed different levels of correlation. Specifically, the strength of correlation between myoelectric activities and principal accelerations (PAk) differs between muscles, indicating that specific muscles play an important role in specific movement components. I added this information as the suggested future research study.

Point 3: 2) The scores rely on the calculation from double differentiation of data gathered from the Kinect V2 sensor. In the previous work [12] motion data were gathered at 250Hz and then filtered at 10Hz, then double differentiated. Are kinematic data gathered in this new way comparable with the ones of the previous study?

Response 3: Thank you very much for pointing out the unclear sentences. In the current study, I used a Fourier analysis to find the suitable cut-off frequency for the current datasets derived from the Kinect V2 sensor. I added more information about this missing step in the revised manuscript and relocated the location of the reference to prevent misunderstanding by the audience.

Point 4: 3) There are many normalization passages that may change the differences between genders. More details about mass distribution normalization should be provided

Response 4: Thank you very much for pointing out the unclear sentences. I rewrote the sub-section of the Methods explaining the pre-processing steps in the revised manuscript to make it clearer than the old one.

Reviewer 2 Report

Dear Authors

Thank you for this manuscript and also "thank you" to the editor and journal for the opportunity to review this article.

For me, it was nice to read this paper.

I have only some minor comments which could maybe improve the manuscript if being addressed:

(1) No abbreviations in the abstract: What are „PP“ and „PA“?

(2) Lines 96/97: I do not unterstand this sentence: „resampled data“?

(3) Lines 115-118: How did you deal with the dependence in the input data since you have 30 x 30 x 1 = 900 rows for each subject?

(4) 2.4. Statistical analysis: Would it be possible to use one kind of statistical test (i.e. non-parametric) for consistency?

(5) 3. Results: Wrong place for indices: It should be „PP3_rVAR“ instead of „PP_rVAR3“ (also in the following cases).

(6) 3: Results: There is a discrepancy in the p-value and effect size for PA3_N between text and table 3.

(7) Maybe it would be helpful for the reader if you can provide a categorisation of the effect size, i.e. „weak“, „medium“ and „strong“, for example.

Especially the point with the dependent data for PCA is important to think about!

Much success for resubmission!

Author Response

Response to Reviewer 2 Comments

Point 1: Thank you for this manuscript and also “thank you” to the editor and journal for the opportunity to review this article. For me, it was nice to read this paper. I have only some minor comments which could maybe improve the manuscript if being addressed:

Response 1: Thank you very much for your valuable time reviewing the manuscript and providing constructive comments and suggestions. I appreciate and apply all the suggestions to improve the manuscript. Please find my changes in the revised manuscript and brief changes in the point-by-point responses below.

Point 2: (1) No abbreviations in the abstract: What are „PP“ and „PA“?

Response 2: Thank you very much for pointing out my mistakes. I removed the words “PP” and “PA” from the Abstract because of the limited number of word counts. I explain these variables in the Methods.

Point 3: (2) Lines 96/97: I do not unterstand this sentence: „resampled data“?

Response 3: Thank you for pointing out the unclear sentences. The word “resampled data” in the old manuscript means that I resampled the sampling rate of the signals to 30 Hz. I rewrote the sentences explaining this step in the revised manuscript and hope it might be more precise than the old one.

Point 4: (3) Lines 115-118: How did you deal with the dependence in the input data since you have 30 x 30 x 1 = 900 rows for each subject?

Response 4: Again, thank you for pointing out the unclear sentences. In order to prepare the data, individual datasets were pre-processing centered, normalized, and weighted (More details of these three pre-processing steps are described in the manuscript). Then, the weighted posture vectors of all participants were concatenated to form one input matric data (104 subjects x 30 seconds x 30 sampling rate (Hz) = 93,600 rows) for PCA analysis. This step allows for comparing the PCA-based variables between the subjects (e.g., between males and females) since they share the total variance. I rewrote this part in the revised manuscript and hoped it might be clearer than the old one.

Point 5: (4) 2.4. Statistical analysis: Would it be possible to use one kind of statistical test (i.e. non-parametric) for consistency?

Response 5: Thank you very much for the suggestion. I used one kind of statistical test for the variable “rVAR” and corrected their results in the manuscript accordingly.

Point 6: (5) 3. Results: Wrong place for indices: It should be „PP3_rVAR“ instead of „PP_rVAR3“ (also in the following cases).

Response 6: I apologize for the mistakes, and thank you for pointing them out. I checked and corrected the mistake in the manuscript accordingly.

Point 7: (6) 3: Results: There is a discrepancy in the p-value and effect size for PA3_N between text and table 3.

Response 7: It is my mistakes. I checked the results and corrected the mistakes already. Thank you very much.

Point 8: (7) Maybe it would be helpful for the reader if you can provide a categorisation of the effect size, i.e. „weak“, „medium“ and „strong“, for example.

Response 8: Thank you for the suggestion. I added more information about the categorization in the Results accordingly.

Point 9: Especially the point with the dependent data for PCA is important to think about!

Much success for resubmission!

Response 9: I carefully check all the dependent data in the manuscript. Thank you very much for your valuable suggestions.